# Rice 3D chromatin structure correlates with sequence variation and meiotic recombination rate

Agnieszka A. Golicz[1 ✉], Prem L. Bhalla[1], David Edwards [2] & Mohan B. Singh[1]

Genomes of many eukaryotic species have a defined three-dimensional architecture critical for cellular processes. They are partitioned into topologically associated domains (TADs), defined as regions of high chromatin inter-connectivity. While TADs are not a prominent feature of *A. thaliana* genome organization, they have been reported for other plants including rice, maize, tomato and cotton and for which TAD formation appears to be linked to transcription and chromatin epigenetic status. Here we show that in the rice genome, sequence variation and meiotic recombination rate correlate with the 3D genome structure. TADs display increased SNP and SV density and higher recombination rate compared to inter-TAD regions. We associate the observed differences with the TAD epigenetic landscape, TE composition and an increased incidence of meiotic crossovers.

---

[1] School of Agriculture and Food, Faculty of Veterinary and Agricultural Sciences, The University of Melbourne, Parkville, VIC 3010, Australia. [2] School of Biological Sciences and Institute of Agriculture, The University of Western Australia, Perth, WA 6009, Australia. ✉email: agnieszka.golicz@unimelb.edu.au

The mechanism and functional significance of DNA packaging in the nucleus has been a long-standing question in biology. The genomes of animals and plants are partitioned into chromatin domains, referred to as topologically associated domains (TADs)[1,2]. These are structural units of chromosome compartmentalization, which emerged as a key feature of higher-order genome organization. They define the regulatory landscapes of chromosomes, form units of co-regulated genes and confine the effect of distal regulatory elements[3]. The abundance of plant TADs is related to genome size; they are not prominent in the compact genome of *Arabidopsis*[3–6], but more abundant in the larger genomes of rice, maize, tomato, sorghum, foxtail millet and cotton[7–9]. Rice TAD borders were reported to be associated with active epigenetic marks and high levels of transcription. Previous results also hinted at an asymmetric epigenetic mark and gene-density distribution across TAD borders[7].

To broaden our understanding of plant TADs, here we asked whether the presence of TADs in the rice genome is associated with changes in nucleotide and structural variant density, using data from the 3000 Rice Genomes Project, the largest publicly available crop plant genome resequencing dataset, as well as the associated publicly available SNP and structural variant calls[10–12]. We found that the rice genome can be divided in TAD and inter-TAD regions. Compared to inter-TADs, TADs have increased sequence variant density, meiotic recombination rate, are over-represented in transposable elements (TEs) and silencing epigenetic marks. Genes found in TADs are shorter, have lower expression levels and are overrepresented in functions related to signalling and response to environmental stimuli.

## Results

**TAD discovery**. To date three Hi-C-based studies of the 3D conformation of the rice genome have been performed[7,8,13]. All three studies identified TADs, however, there were substantial differences in domain size, genome coverage by the domains identified as well as intra-domain interaction strength (Supplementary note, Supplementary Table 1). The differences most likely reflect the TAD discovery algorithms used as well as the underlying TAD definitions adopted. Liu et al.[7] defined TADs only as regions of very strong interaction signals and the resulting TADs identified were relatively small (median size 45 to 50 kb) covering about a third of the genome, whereas TADs identified by Dong et al.[8,13] were much larger (median size of 160 kb and 450 kb, respectively) covering higher proportion of the genome, but also had much lower median intra-TAD interaction signal (Supplementary Table 1). Together these results suggest that rice TADs, similar to what was found in metazoans[14], could have hierarchical structure with smaller, densely interacting domains contained within larger scale domains of hundreds of kilo-base pairs. However, the analysis performed by Liu et al.[7], as well as visual inspection of the interaction maps of rice genome (Fig. 1a), clearly indicates that there exist smaller regions of increased interactions, which manifest as strong triangular Hi-C signals[15]. We were intrigued by those and wondered whether any sequence features distinguish those from the remainder of the rice genome. We therefore used Armatus[14], a software package designed specifically for the discovery of densely interacting domains and which has been shown to outperform both Arrowhead[16] and DomainCaller[17] algorithms in benchmarking comparisons[18]. We performed TAD discovery using all three rice Hi-C datasets available[7,8,13], however, ultimately chose TADs identified by Armatus from Liu et al.[7] dataset for final analysis, based on biological replicate concordance, number of valid pairs identified by Hi-C-Pro, TAD size and interaction strength (Supplementary note). We then compared the coordinates of 4599 TADs

identified in our study with those reported by Liu et al[7] and found that both sets of TADs overlap significantly more than it would be expected by chance alone (Fig. 1b, Supplementary data 1). We also investigated the distribution of epigenetic marks at TAD boundaries and found prominent H3K4me3, H3K9ac and H4K12ac peaks centred on TAD boundaries (Fig. 2, Supplementary Fig. 1), which is consistent with previous findings of enrichment of those epigenetic marks at rice TAD boundaries[7,8,13]. The TADs discovered had a median size of 35 kb and covered 69.7% of the rice genome, while regions falling outside of identified TADs (inter-TADs) had a median size of 25 kb and covered 30.3% of the rice genome.

We then set out to explore whether there exist any genetic and epigenetic features which distinguish the densely interacting TADs (regions identified as TADs by Armatus) from inter-TAD regions (genomic regions falling outside of TADs identified by Armatus) (Fig. 1a).

**Different variation profiles of rice and human TAD borders**. A significantly reduced sequence variation is observed at TAD borders within the human genome, and this sequence conservation has been attributed to the requirement for boundary formation and recognition by boundary-binding proteins, including CTCF[19]. Deletion and mutation of TAD borders has in turn been linked to pathogenicity, including developmental disorders and cancers[19,20]. Although no CTCF homologue had been found in plants, homologues of cohesins are present[2,21]. One of the important outstanding questions regarding TAD formation in plants is whether sequence-specific boundary recognition is necessary for TAD formation. Genome-wide analysis of variant density allowed us to compare plant and human TAD boundary variant profiles.

Conservation of human TAD boundaries is reflected in SNP and structural variant distribution across the human genome[19]. When SNP and structural variant breakpoint density is plotted across TAD borders, there is a pronounced reduction in density exactly at the TAD border[19] (Fig. 2a). Human TAD borders also display increased gene density[17], which likely contributes to the variation profile observed (Fig. 2a). However, upon partitioning into genic and intergenic sequence the trend of reduced sequence variation at TAD boundary persists, supporting importance of boundary sequence conservation (Supplementary Fig. 2). We then employed the same apprach[19] to study variant profile of TAD borders across the rice genome. We would expect that if there were a requirement for sequence recognition for boundary formation in rice, a similar reduction in variant density at TAD border would be found. However, in rice no immediate reduction in variant density at TAD boundary was observed (Fig. 2b). We did observe a reduction in variant density at ~5 kb before TAD boundary, but this most likely corresponds to an increase in gene density, as in contrast to what is observed in human, the 5 kb dip in variant density disappears upon partitioning into genic and intergenic sequence (Supplementary Fig. 2). The finding suggests that sequence conservation at TAD boundaries may be less important for rice than it is for human TAD formation. In addition, we noted an overall uneven distribution of both SNPs and SV breakpoints between TADs and inter-TADs, with TADs appearing to have a higher density of genomic variants.

**Rice TADs have higher variant density compared to inter-TADs**. Having observed an asymmetry in variant distribution across TAD borders (Fig. 2), we then turned to comparisons of entire TAD and inter-TAD regions. First, we asked if TADs and inter-TADs differ in genomic variant distribution. We investigated the distribution of 6,495,641 SNPs, 3,534,001 deletion and 410,823 insertion breakpoints in the rice genome. We

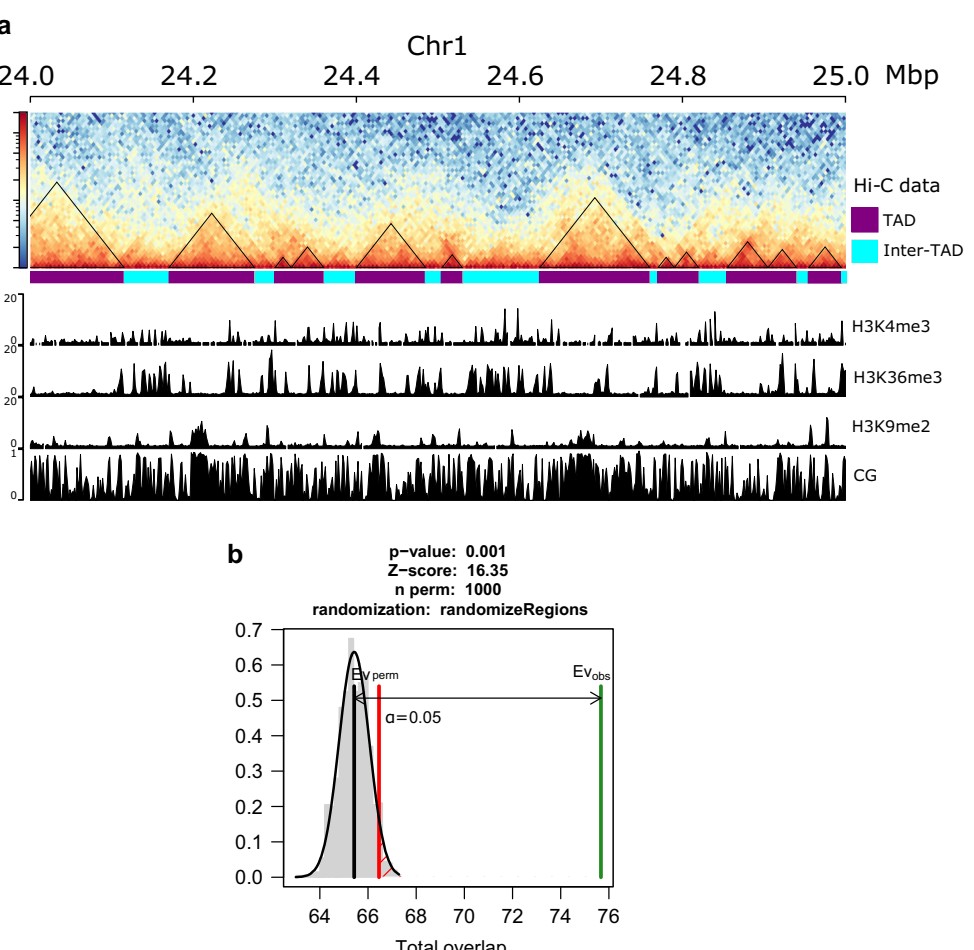

**Fig. 1 Distribution of TADs across the rice genome. a** Contact maps showing TADs across a section of rice chromosome 1. TADs identified as regions of high chromatin interactions by Armatus are outlined by black triangles. The remaining regions are inter-TADs. Tracks below Hi-C data present selected epigenetic modifications in TAD and inter-TAD regions. **b** Overlap between TADs called by Armatus in this study and those identified by Liu et al.[7] Both sets of TADs overlap much more than it would be expected by chance alone. Green line—observed overlap, grey line—mean of simulated overlap, red line —significance threshold. TADs: $n = 4599$, inter-TADs: $n = 3346$.

found that both SNP and structural variant breakpoint density is higher within rice TADs compared inter-TADs (Fig. 3). Rice TADs and inter-TAD regions differ in protein coding gene density (median TAD gene density—0.11 gene/kb, median inter-TAD gene-density—0.15 gene/kb, two-tailed Wilcoxon rank sum test $p < 0.001$) and gene bodies demonstrate reduced SNP and SV abundance compared to intergenic regions due to functional constraints[11,12]. To avoid bias due to differential gene density between regions, we partitioned the rice genome into exons, introns and intergenic sequence. Comparison of these domains demonstrated an increased SNP and insertion and deletion breakpoint density within TADs across both genic and intergenic sequences (Fig. 3). The results suggest that some properties of TADs, beyond differences in gene density, may contribute to their increased sequence variation.

**Correlation between rice genomic and epigenomic features**. We wanted to investigate the potential causes for the differences in SNP and SV breakpoint density between TAD and inter-TAD regions. To that end we first performed a genome-wide analysis attempting to correlate different sequence and epigenetic features across the rice genome. A similar strategy has previously been used to for the analysis of human data and revealed that the chromatin state is a major influence on regional mutation rates[22]. For example, in human, H3K9 di- and tri- methylation and

methylation in CG context have been associated with increased SNV mutation rate[22–25]. In medaka, a Japanese rice fish, DNA hypermethylation is associated with an increase in all types of single nucleotide substitutions, not just spontaneous oxidative de-amination of methylated CpG[24]. Some of the proposed mechanisms of action include alternative DNA repair pathways and impaired access of repair machinery due to chromatin compaction[24,26]. Similarly, our genome-wide analysis revealed significant positive correlation between SNP density across both genic and intergenic regions, CG and CHG DNA methylation and H3K9 di-methylation (Fig. 4, Supplementary Figs 3 and 4, Supplementary data 2). SNP density in both genic and intergenic regions was also positively correlated with transposable element density (especially CACTA DNA transposons and retro-transposons) revealing potential link between transposon density, heterochromatin and SNP accumulation. Transposons are subject to heterochromatic silencing associated with high levels of CG and CHG DNA methylation[27], and heterochromatic silencing has been shown to spread to the surrounding sequences in maize[28] and rice[29]. As transposon density as well as CG, CHG and H3K9me2 methylation are positively correlated with SNP density, heterochromatic spreading could contribute to SNP accumulation. Condensed chromatin self-organization[30,31] could also impair access of the DNA repair machinery. In addition, compared with RNA transposons, DNA transposons had stronger

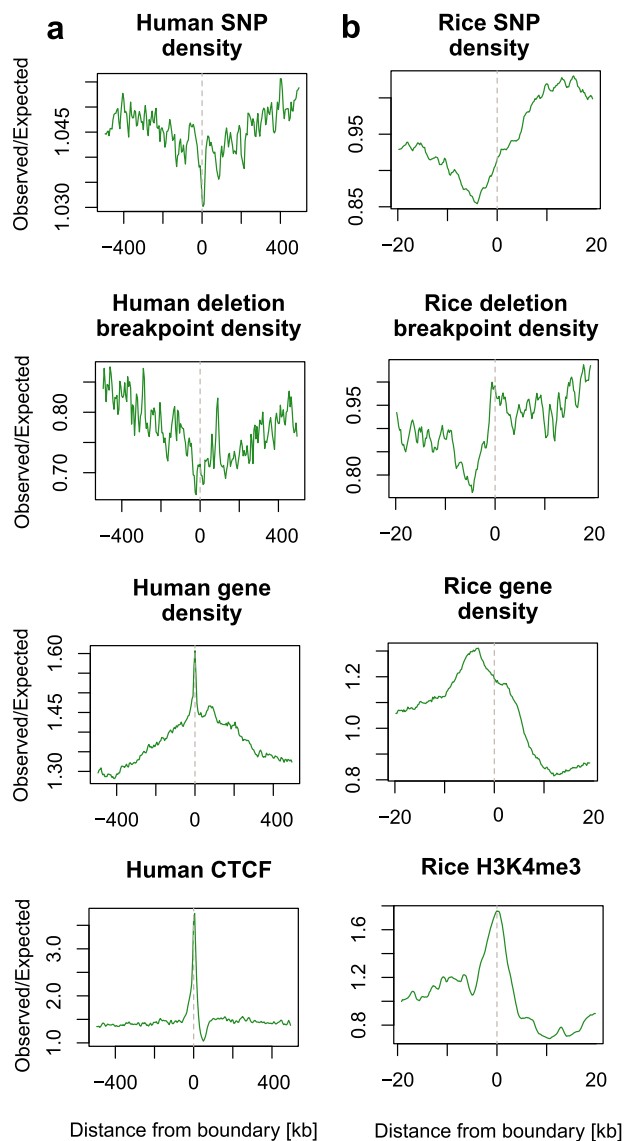

**Fig. 2 Variant profiles across human and rice TAD boundaries. a** Human variant profile across TAD boundaries. **b** Rice variant profile across TAD boundaries. For ease of comparisons with published literature following conventions of Liu et al.[7] and Fudenberg et al.[19] 20 kb to the left and right of the boundary were shown for rice and 500 kb for human.

positive correlation with deletion breakpoints, while RNA transposons were more strongly correlated with insertion breakpoints, in accordance with the postulated transposon origin of many reported rice SVs[12]. Finally, we observed an association between variant density and recombination rate, especially SNP density being positively correlated with recombination rate. The observation is in line with previous findings as increased SNP density was found to be associated with meiotic crossovers[32–34].

We then set out to test which of the features identified in this genome-wide analysis could contribute to increased variant density within TADs.

**TADs and inter-TADs differ in epigenetic marks, TEs and recombination rate.** First, we compared levels of DNA methylation and histone modifications between TADs and inter-TAD regions. We found that TADs are lower in active and higher in repressive epigenetic marks compared to inter-TAD regions (Fig. 5a). TADs have a significantly higher CG and CHG DNA methylation and H3K9me2 levels than inter-TADs, a feature which has been shown to be mutagenic in humans[22] and positively correlated with SNP density in our genome-wide analysis, suggesting that it could contribute to single nucleotide mutation accumulation in TADs. The results are also consistent with asymmetry in epigenetic marks across TAD boundaries observed (Supplementary Fig. 1) and previously reported[7]. We also observed that the proportion of C to T nucleotide substitution is higher in TADs than in inter-TADs (median fraction for TADs 0.2419, median fraction for inter-TADs 0.2277, two-tailed Wilcoxon rank sum test $p < 0.001$), suggesting that spontaneous oxidative de-amination of CpG may also add to the observed differences in sequence variation between TAD and inter-TAD regions. The findings suggest that chromatin state and epigenetic modifications may contribute to the differential single nucleotide variant accumulation across TAD and inter-TAD regions.

Increased presence of silencing marks is often associated with transposable elements (TEs)[35] and our genome-wide analysis pointed to correlation between TEs, epigenetic silencing marks and variant density. We therefore investigated the TE composition of TADs and found differential TE partitioning between TAD and inter-TAD regions. We found that TADs are overrepresented in CACTA DNA transposons and retrotransposons (Fig. 5b), which have been found to be positively correlated with SNP density (Fig. 4). Increased TE density in TADs may promote heterochromatic silencing of the surrounding chromatin[27,28] or condensed chromatin self-assembly[30,31], which in turn could contribute to single nucleotide variant accumulation. In addition TEs in rice are

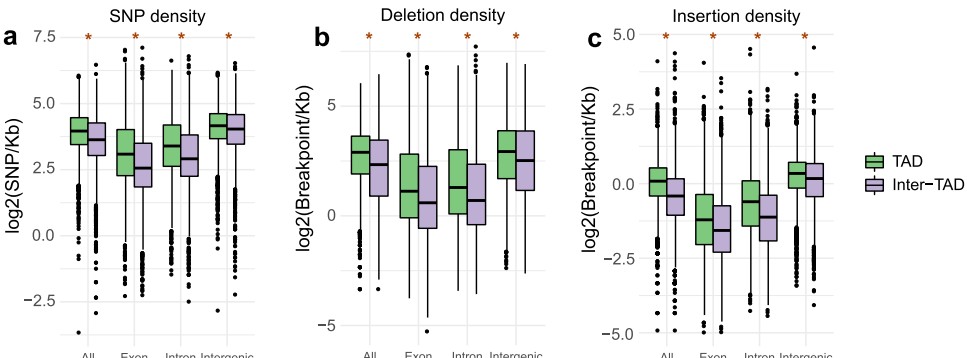

**Fig. 3 Variant density in TADs and inter-TAD regions. a** SNP, **b** deletion breakpoint and **c** insertion breakpoint density is significantly higher in TADs compared to inter-TAD regions. *$p < 0.01$, two-tailed Wilcoxon rank sum test, statistical tests were performed on non-log-transformed values. centre line, median; box limits, upper and lower quartiles; whiskers, 1.5× interquartile range; points, outliers.

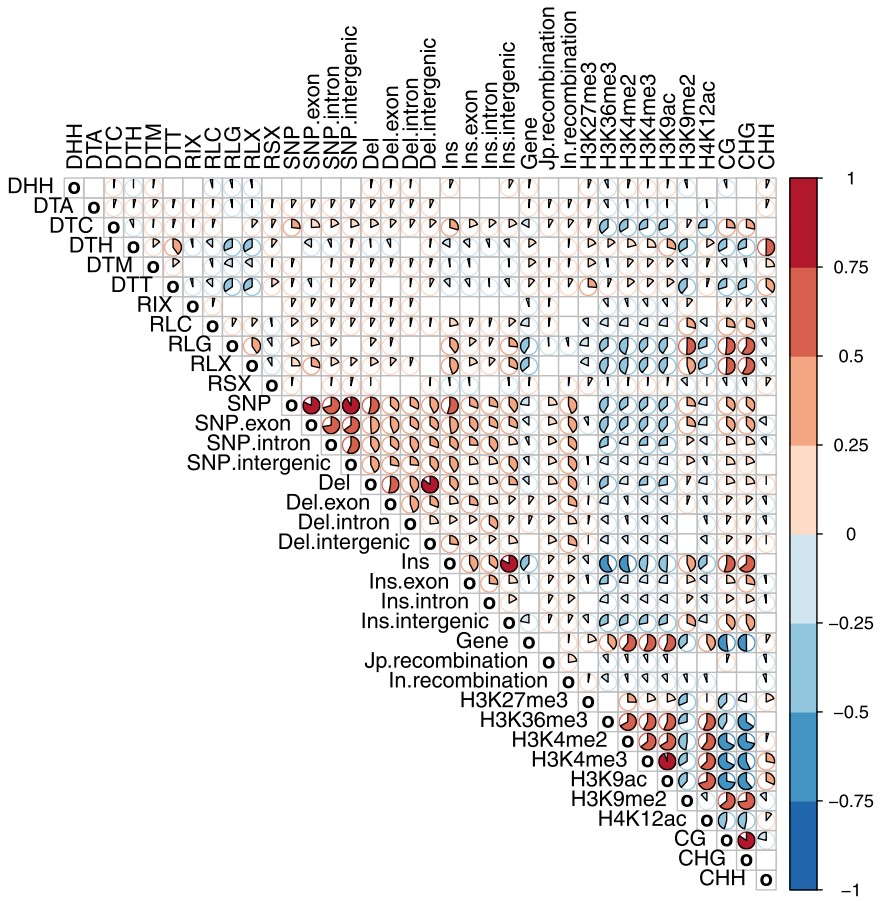

**Fig. 4 Correlations between different genomic and epigenomic features across the rice genome.** The genome was split into 40 kb bins for the analysis. Individual correlation plots can be obtained from Supplementary data 2. Red colour—positive correlation. Blue colour—negative correlation. Pie fill size is proportional to correlation. Plot presents Pearson correlations computed for log2-transformed values (Pearson and Spearman correlations for non-log-transformed values are shown in Supplementary Figs 3 and 4). Only significant correlations (cor.mtest implemented in package corrplot, $p < 0.05$) are shown. Del deletion, Ins insertion, Jp japonica, In indica. DHH—Helitron, DTA—hAT, DTC—CACTA, DTH—PIF-Harbinger, DTM—Mutator, DTT—Tc1-Mariner, RIX—LINE, RLC—LTR Copia, RLG—LTR Gypsy, RLX—LTR Superfamily unknown, RSX—SINE.

associated with insertions and deletions[12] (Fig. 4) and increased TE density in TADs most likely at least partially explains increased density of SV breakpoints.

Increased SNP density was previously found to be associated with meiotic crossovers[32–34]. Our genome-wide analysis also revealed significant positive correlation between SNP density and meiotic recombination rate (Fig. 4). We investigated whether TADs have a higher recombination rate and are more likely to overlap meiotic crossover (CO) events. We used two publicly available datasets, including 1067 unique breakpoints generated by resequencing of 38 rice F2 *indica* individuals[33] and genome-wide recombination rate identified from 75 *indica* and 75 *japonica* accessions from the 3000 Rice Genomes Project[34]. We found that TADs have significantly higher recombination rate compared to inter-TAD regions (Fig. 5c). TADs overlapped more breakpoints than would be expected by chance (Fig. 5d). Interestingly, meiotic crossovers are also known to be associated with open chromatin state and high levels of H3K4me3, H3K9ac and H4K12ac[34], which is decreased in TADs, compared to inter-TAD regions. On the other hand, retrotransposon methylation was shown to be positively correlated with recombination rate[29]. A mechanism integrating multiple contributing factors including the underlying genome sequence, epigenetic modifications and 3D chromatin structure likely controls meiotic crossover positioning and recombination rate.

**Genes found in TADs and inter-TADs have different functions.** Our analysis suggests that the genome is effectively partitioned into TADs overrepresented in silent chromatin marks and transposable elements and inter-TAD regions associated with active chromatin. We were curious if the genes found within TADs and outside of TAD regions differ in any other properties beyond variant density. We found that genes found in inter-TAD regions are on average longer (Fig. 6a) and expressed at higher levels (Fig. 6b). We also searched for functional overrepresentation of TAD and inter-TAD genes. We found the inter-TAD genes to be overrepresented in functions related to translation, oxidative phosphorylation and protein transport (Fig. 6d) and the TAD genes to be overrepresented in functions related to protein phosphorylation and regulation of gene expression (Fig. 6c). The inter-TAD genes appear overrepresented in house-keeping functions while the TAD genes are overrepresented with functions related to signalling and responses to environmental stimuli.

## Discussion

Our observations mirror that made in *Drosophila melanogaster*, where a proposed model suggests that chromatin is divided in TADs and inter-TAD regions[31]. With inter-TADs harbouring active chromatin and genes overrepresented in house-keeping functions. Our observations point to a similar organization of the

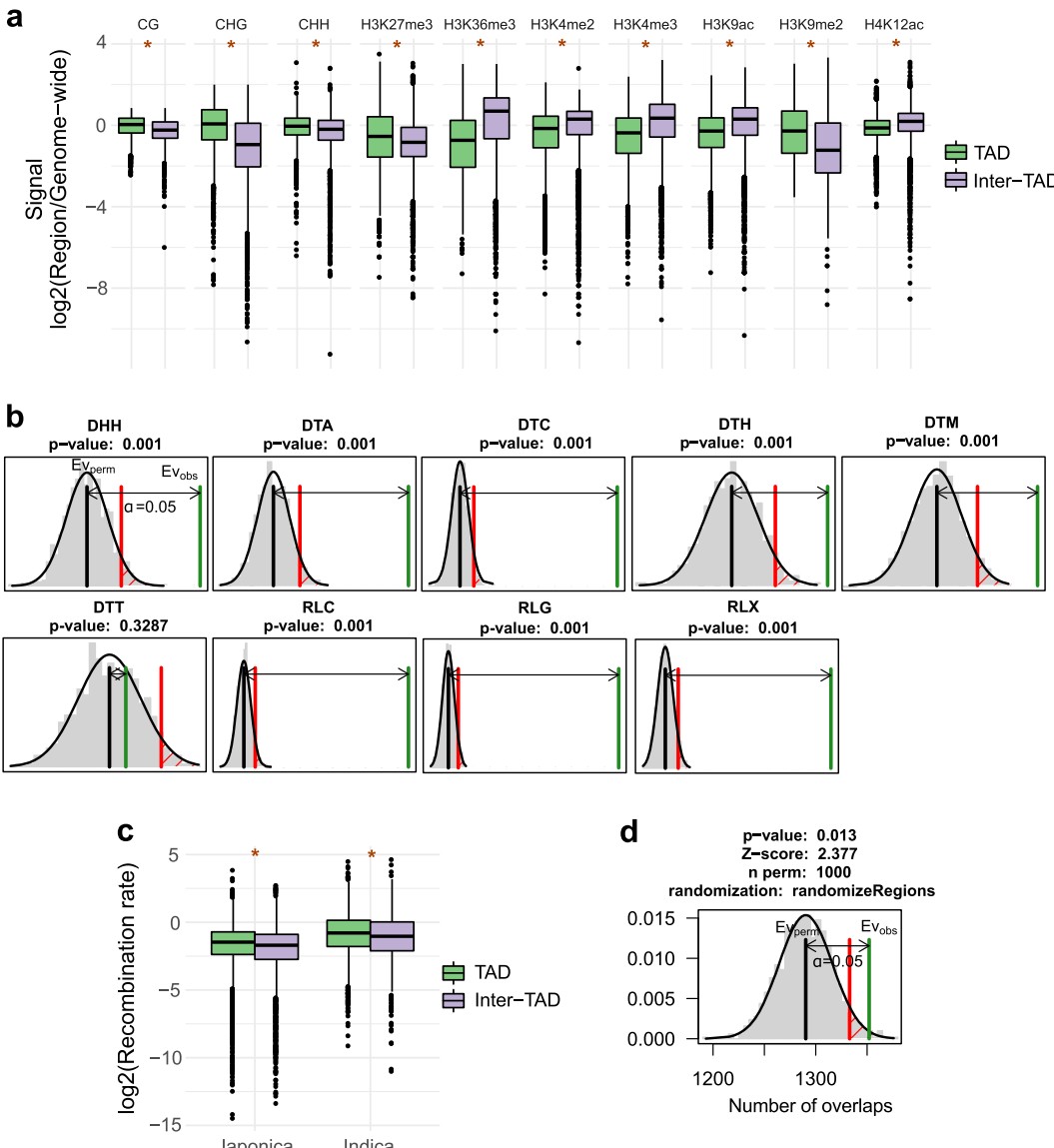

**Fig. 5 Comparison of genomic and epigenomic features between TAD and inter-TAD regions. a** Histone and DNA modifications. TADs are higher in repressive and lower in active chromatin marks compared to inter-TAD regions. **b** Transposable element (TE) permutation analysis. Most TE classes are overrepresented in TADs. Green line—observed number of overlaps, grey line—mean of simulated overlaps, red line—significance threshold. Thousand permutations were performed. **c** Recombination rate. Recombination rate is significantly higher in TADs compared to inter-TAD regions. **d** Number of meiotic crossovers. Meiotic crossovers are significantly overrepresented in TADs. Green line—observed number of overlaps, grey line—mean of simulated overlaps, red line—significance threshold. *$p < 0.01$, two-tailed Wilcoxon rank sum test, statistical tests were performed on non-log-transformed values. Center line, median; box limits, upper and lower quartiles; whiskers, 1.5× interquartile range; points, outliers.

rice genome, which is also reminiscent of the compacted and loose structural domains identified in Arabidopsis, albeit at a smaller scale[3–5]. We also demonstrate that in contrast to human TAD borders, rice TAD borders do not show signatures of purifying selection, supporting the hypothesis that sequence recognition by insulating proteins in not necessary for TAD formation and consistent with the *Drosophila* model of TAD self-assembly due to an ability of inactive chromatin to aggregate[31]. We further extend those findings by analysis of sequence variants and find that TADs have a higher density of variants across both coding and non-coding regions. Our results suggest that beyond involvement in chromatin folding heterochromatic TADs may form a mutagenic environment which could contribute to variant accumulation, as increased SNP density was observed both across genic and non-genic regions.

## Methods

**TAD identification.** The three existing rice Hi-C datasets were obtained from Sequence Read Archive (PRJNA391551, PRJNA354683, PRJNA429927)[7,8,13]. Interaction maps were built using Hi-C-Pro[36] v2.11.1 at 5 kb pair resolution using rice genome MSU7/IRGSP1.0[37] as a reference. Interaction maps were first built separately for the biological replicates to investigate concordance between results obtained (Supplementary note) and biological replicates were then merged to obtain final interaction maps used for TAD identification. TADs were identified using Armatus[14] v2.3. Several values of the γ parameter which controls TAD size were tested and the final value used was 0.4 (Supplementary note, Supplementary Table 2 and Supplementary Table 3).

**Rice SNP and SV analysis.** Rice genome and genome annotation (MSU7/IRGSP1.0) were downloaded from Phytozome v12[38]. SNP (3K RG 18mio base SNP Dataset) and SV (RG Large Structural Variants release 1.0) datasets were downloaded from SNP-Seek database[39]. SNPs were filtered to include only those found in Japonica and Indica lines with a minimum minor allele frequency of 0.01 using VCFtools[40]. SV variants were pre-filtered to include only those found in Japonica

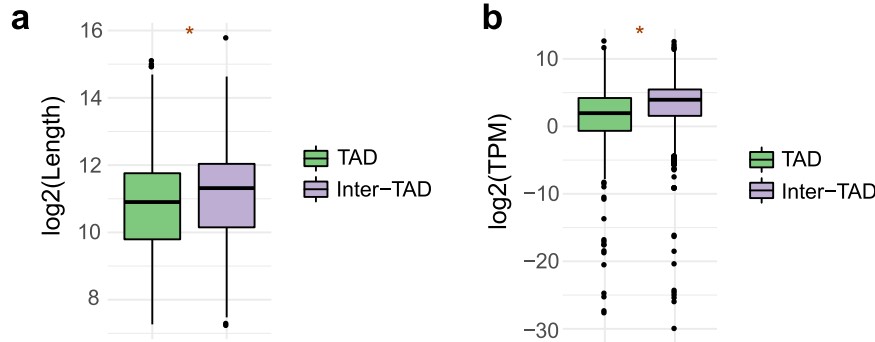

**c**

| GO ID | Term | Annotated | Significant | P value |
|---|---|---|---|---|
| GO:0006468 | Protein phosphorylation | 1558 | 1008 | 2.20E-15 |
| GO:0043687 | Post-translational protein modification | 1750 | 1106 | 3.20E-12 |
| GO:0006915 | Apoptotic process | 465 | 327 | 1.40E-11 |
| GO:0006952 | Defense response | 435 | 300 | 2.70E-09 |
| GO:0071554 | Cell wall organization or biogenesis | 138 | 113 | 1.20E-08 |
| GO:0006857 | Oligopeptide transport | 99 | 80 | 8.00E-08 |
| GO:0010468 | Regulation of gene expression | 1562 | 962 | 8.50E-07 |
| GO:0019219 | Regulation of nucleobase-containing compound | 1501 | 922 | 2.00E-06 |
| GO:0048544 | Recognition of pollen | 95 | 74 | 3.50E-06 |
| GO:0009889 | Regulation of biosynthetic process | 1539 | 947 | 5.40E-06 |

**d**

| GO ID | Term | Annotated | Significant | P value |
|---|---|---|---|---|
| GO:0006412 | Translation | 682 | 314 | 3.90E-24 |
| GO:0006119 | Oxidative phosphorylation | 89 | 57 | 1.50E-12 |
| GO:0006886 | Intracellular protein transport | 206 | 96 | 2.30E-08 |
| GO:0016192 | Vesicle-mediated transport | 192 | 88 | 6.80E-07 |
| GO:0015986 | ATP synthesis coupled proton transport | 55 | 33 | 6.90E-07 |
| GO:0065008 | Regulation of biological quality | 192 | 81 | 4.80E-06 |
| GO:0042180 | Cellular ketone metabolic process | 437 | 165 | 5.40E-06 |
| GO:0022607 | Cellular component assembly | 239 | 111 | 5.90E-06 |
| GO:0051186 | Cofactor metabolic process | 224 | 95 | 6.60E-06 |
| GO:0044283 | Small molecule biosynthetic process | 364 | 145 | 1.10E-05 |

**Fig. 6 Genes in TAD and inter-TAD regions are significantly overrepresented in different functional categories. a** Gene length in TAD and inter-TAD regions. **b** Expression levels in TAD and inter-TAD regions. Genes within TADs have significantly lower expression levels. **c** GO terms overrepresented among genes found in TADs. **d** GO terms overrepresented among genes found in inter-TAD regions. *$p < 0.01$, two-tailed Wilcoxon rank sum test, statistical tests were performed on non-log-transformed values. Center line, median; box limits, upper and lower quartiles; whiskers, 1.5× interquartile range; points, outliers.

and Indica lines. Filtering was performed for consistency with Hi-C data (Japonica lines only) and recombination data (Japonica and Indica lines) available. For deletions, start and end coordinates were used as breakpoints. For insertions, start coordinates were used as breakpoints.

**Rice recombination, epigenetic and expression data**. Crossover breakpoints were obtained from ref. [33]. Recombination rates across the rice genome were obtained from ref. [34]. Histone modification ChIP-Seq data were obtained from PlantDHS[41]. DNA methylation data were obtained from MethBank[42] v3.0. Gene expression data were obtained from ref. [7] (PRJNA354683). Expression levels were quantified using Kallisto[43] v0.45.0.

**Rice TE annotation**. The non-redundant set of TEs was obtained from TREP database[44]. RepeatMasker v4.0.7, with default parameters using TREP database, was used to annotate the TEs in the rice genome.

**Overlap analysis**. The overlaps between TADs, genes, variants, histone and DNA modifications were computed using bedtools[45] v2.28. The significance of overlap between Liu et al. TADs, crossover breakpoints, TEs and Armatus TADs was evaluated using regioneR[46] v1.15.2 overlapPermTest function by random region re-distribution across the genome (randomizeRegions).

**Relative abundance at TAD boundary**. Relative abundance calculation was performed based on the score introduced by Fudenberg et al.[19] with a minor

adjustment of operating on densities rather than total counts and using a random re-distribution of windows rather than genome-wide abundance to obtain expected values to account for potential bias resulting from performing calculations on genomic windows.

Re-distribution of TAD boundaries across the rice genome was performed using randomizeRegions function of regioneR v1.15.2.

**Genome-wide feature correlation analysis**. The genome was split in 40 kb non-overlapping windows. Densities and signal levels for the corresponding genomic and epigenomic features were computed using bedtools[45] v2.28 intersect and map functions. Pearson correlations were computed for log2-transformed and non-transformed values. Spearman correlations were computed for non-transformed values. Significance computed using cor.mtest implemented in package corrplot was used for significance testing ($p < 0.05$). To avoid log2(0), 0.1 (median value of the entire dataset) was added to all values.

**Gene ontology (GO) analysis**. GO annotation was obtained from Agrigo2[47]. TopGO[48] v2.35.0 was used to compute functional overrepresentation of genes using method 'weight' to adjust for multiple comparisons.

**Human genomic diversity analysis**. Human 1000 genomes variants[49] were downloaded from ENSEMBL and pre-filtered retaining only single nucleotide variants (SNVs) with minor allele frequency of over 0.01. Coordinates of human TADs were obtained from[16] (GSE63525). Human deletions were obtained from[50] (dbVar nstd100).

**Statistics and reproducibility**. Differences between TADs and inter-TADs were tested for statistical significance using two-tailed Wilcoxon rank sum test implemented in R. Overrepresentation of features within TADs was tested for statistical significance using permutation test implemented in regioneR. Pearson and Spearman correlations were computed using cor function in R and significance was computed using cor.mtest function implemented in the package corrplot. Enrichment of GO terms was analyzed suing topGO package, using method 'weight' to adjust for multiple comparisons.

**Reporting summary**. Further information on research design is available in the Nature Research Reporting Summary linked to this article.

## Data availability

All data used in the manuscript is publicly available (See "Methods").
Supplementary data 1 and 2 can be found under https://osf.io/hevzb/.

## Code availability

No custom code or mathematical algorithm central to the conclusions was used.

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

## Acknowledgements
We would like to thank Dr Philipp Bayer for the valuable comments on the manuscript. This work was supported the University of Melbourne McKenzie Fellowship.

## Author contributions
AAG conceived the study, designed the experiments, performed the analysis and wrote the manuscript, PLB and MBS edited the manuscript, DE provided critical comments on the study design and edited the manuscript. This research was supported by Spartan HPC at the University of Melbourne, Australia.

## Competing interests
Authors declare no competing interests.
