## [Peer review file · Communications Biology]

Reviewers' comments:

Reviewer #1 (Remarks to the Author):

This manuscript takes effort in looking at the genetic variations associated with the pattern of TAD in rice, using published data of TAD information, published data of genome-wide genetic variation and published pattern of genetic recombination in rice. The conclusion made from this analysis suggested that there might be more genetic variation in the boundaries of TAD in rice as well as higher rate of genetic recombination, a result that is different than what was observed in human.

I have some doubt about the overall conclusion of this manuscript, which also might due to the method/data they used. Below are my specific concerns:

1. To make a fair comparison, the identification of TAD or non-TAD is very critical. The authors has simply used the TAD information identified in the Nature Plant paper that was filtered by what was called arrowhead method, which can missed quite some less obvious TAD. I would suggest to use the TAD information that was identified in the MP,TPJ paper, in addition to that of the Nature Plant paper.
2. Fig1B, the SNP density inside the TAD looked like still somewhat higher than that of the TAD boundary, a result that is quite consistant with what reproted in animal system? The same goes to the insertion analysis.
- 3.It would be good to indicate how the "random" control was done in you method.

Reviewer #2 (Remarks to the Author):

This manuscript performed a set of computational analysis integrating existing genomic resources including rice chromatin conformation map and variants across rice populations. The manuscript described a somewhat interesting observation that nucleotide variations are enriched across TAD borders, which is different from what was observed in the human. However, the manuscript is rather preliminary and does not provide sufficient novelty or biological insights into the described observation. The results are also rather fragmented and do not provide a convincing and coherent explanation of the rice-human difference. The enrichment of meiotic crossover (CO) at TADs appears to be interesting. However, a rice-human comparison for CO was not performed and hence it is unclear whether the enrichment of CO contributes the rice-human difference. Statistical tests are also missing for a number of conclusions in the manuscript.

- 1.Many genomic features being analyzed in this manuscript are not clearly defined. To name a few examples, Line 34-36: It is unclear what the authors mean with "within TADs", or "surrounding genomic sequence". What is the size of these features? Line 36-37: what is the definition of "non-TAD" regions? Does it mean all genomic regions not annotated as TAD, or only surrounding regions of TADs?
- 2.Fig. 1A top two panels: the Y-scales are too compressed that any difference between TAD and non-TAD cannot be seen easily.
3. In order to compare rice to human, the authors should present the human results as part of Fig. 1B. So the reads don't have to go back to published literature about human chromatin organizations.
4. Fig. 1B. For the distribution of deletions and insertions, there is substantial average fluctuation of the densities surrounding TAD boundary. Is the enrichment of deletion and insertion at TAD boundary statistically significant? A shuffling experiment may be performed to test whether the amplitude of increase at TAD boundary is significant.
5. Statistical tests are missing from the following lines - Line 55-56, Line 84-87, Line 90-91,

6. Line 60. How are TAD and non-TAD regions defined here? What are the sizes?

7. The authors should perform rice-human comparison for the TE and meiotic crossover enrichment analyses to ask whether these are possible contributors for the observed rice-human differences.

Reviewer #3 (Remarks to the Author):

This manuscript by Agnieszka et al. asked the question whether presence of topologically associated domains or TADs in the rice genome is associated with reduced single nucleotide polymorphisms (SNPs) and structural variations as previously reported in the human genomes. Authors used the 3000 rice genome resequencing data for this purpose. They found increased SNPs and structural variant density across TAD borders and within TADs in rice, which was in contrast to findings in human genomes. Potential effects of differences in gene density between TADs and non-TADs were ruled out by separately analyzing genic and non-genic loci across the rice genome. Next, authors tried to associate the high rate of genetic variation with the TAD epigenetic landscape, TE composition and meiotic crossover rates. They concluded that epigenetic modifications, partitioning of different TE families as well as meiotic crossing rates may all contribute to the higher level of genetic variation in TADs than non-TADs in rice.

The subject of this manuscript is topical. Overall, the analytical methods are sound and the manuscript is well-written. It provides new insight into the topological chromatin structure and its maintenance in rice genome, although the study is only based on sequence-level correlative analyses and descriptions while lack of any experimental causal validation. It also remains unknown whether trends identified in rice will apply to other plant genomes even within the monocots. For example, if differential presence of TEs is a major contributing factor, then we would expect fundamental differences between rice and maize, barley and wheat. In this aspect, the study and conclusions cannot be considered as generally applicable. It only draws the attention that there is a difference between rice and human genomes with respect to the genetic mutation rate of TADs vs. non-TADs. In addition, some of the conclusions may not have been strongly supported by even by the data and/or analyses. I have the following additional comments.

1. I wonder if the cell-type-specific TADs identified by Dong et al, 2017 (Molecular Plant) is more suited for this purpose than those by Liu et al, 2017 used in this manuscript. TADs identified in the latter paper utilized the rice seedlings as its input tissue. Accordingly, the respective Hi-C interaction map was a weighted average interaction status across all various cell types, although majority should be the mesophyll cells. To avoid potential effects of the cell-type differences and ensure the specificity and accuracy of the utilized TADs, it might be better to use the pure mesophyll cell TADs characterized by Dong et al, 2017, or, to use both.

2. It seems to me that according to presentations in Fig.1B and Fig. S1, except for the SNP density in non-genic regions (Fig. S1 bottom right panel), the densities of "Deletion and Insertion" around the TAD borders and within the TADs (TAD inner regions) are largely (in both genic and non-genic regions) indistinguishable from those generated by random selection (set as the background). Accordingly, at Page 2 lines 54-56, it needs to be further specified as "only SNP density in non-genic region was elevated across TADs (around the TAD borders and within the TAD boundaries).

3. Page 2, line 59, it is indeed difficult to establish a true causal relationship between the elevated sequence variation and the TADs. What can be explore is the association between these two properties, genomic and chromatin architecture.

4. Page 2, lines 57-58, the statement about "the lack of insulator proteins identified in plants" was inaccurate. It is true that there are no CTCF factors in plants; however, isolative cohesins are encoded in plant genomes.

5. Page 3, lines 64-67, I suggest authors emphasize the distinct differences of enrichment in active and repressive histone markers in TAD border and TAD inner regions, respectively. The enrichment of H3K9me2 within TAD could only explain the mutation accumulation in TAD inner regions not that around the TAD borders.

3. Page 5, line 147, it should be "the coordinates of rice TADs were obtained from a previous publication5"?

We appreciate the reviewers' careful reading of the manuscript and their constructive comments. Our responses are presented in the blue text below.

This manuscript takes effort in looking at the genetic variations associated with the pattern of TAD in rice, using published data of TAD information, published data of genome-wide genetic variation and published pattern of genetic recombination in rice. The conclusion made from this analysis suggested that there might be more genetic variation in the boundaries of TAD in rice as well as higher rate of genetic recombination, a result that is different than what was observed in human.

The manuscript reports two sets of results. First, we perform, side by side comparison of human and rice variation profiles at TAD border which suggests that in contrast to the human genome, in the rice genome no pronounced reduction of variant density at TAD boundary is observed (Section titled 'Rice and human TAD borders have different variation profiles' and corresponding Fig 2). Having performed the comparative human-rice analysis we made an interesting observation that in rice the overall variant profile across boundaries appears highly asymmetrical. We then turned to comparison of entire rice TADs and inter-TADs (not just the border regions) and show that overall rice TADs have higher SNP density, SV density and recombination rate compared to inter-TADs (Section titled: 'Rice TADs display increased variant density compared to inter-TADs', 'Rice TAD and inter-TAD regions differ in epigenetic landscape, TE density and meiotic crossover rate' and corresponding Figs 3 and 5). We have re-structured the manuscript to emphasize that the two distinct analyses.

I have some doubt about the overall conclusion of this manuscript, which also might due to the method/data they used. Below are my specific concerns:

1. To make a fair comparison, the identification of TAD or non-TAD is very critical. The authors has simply used the TAD information identified in the Nature Plant paper that was filtered by what was called arrowhead method, which can missed quite some less obvious TAD. I would suggest to use the TAD information that was identified in the MP,TPJ paper, in addition to that of the Nature Plant paper.

There were significant differences between the three studies in the number, genome coverage by the topological domains identified as well as intra-TAD interaction strength, most likely stemming from the different algorithms, filtering parameters used and underlying definition. We now discuss those in the main text (Pages 2-3, Lines 35-68 of the revised manuscript) as well as the Supplementary text. In order to build a uniform framework for the study we now performed TAD discovery for all three datasets using Armatus, which has been shown to outperform both Arrowhead and DomainCaller algorithms used in previous studies¹. We used Armatus because it is known to discover TADs of higher intra-TAD interaction strength compared to DomainCaller. We now make a clear distinction in the manuscript that in our analysis TADs are defined as regions of high interaction strength as identified by Armatus, non-TADs (now termed inter-TAD regions) are regions found outside of TADs. The revised version of the manuscript includes section titled 'TAD discovery' which discusses the above-mentioned issues in more detail.

2. Fig1B, the SNP density inside the TAD looked like still somewhat higher than that of the TAD boundary, a result that is quite consistent with what reported in animal system? The same goes to the insertion analysis.

We now include re-analysis of the human single nucleotide variants (SNV) and deletion breakpoints around TAD border. Side-by-side comparisons reveal that while in human variant density is reduced exactly at TAD border, that feature is not observed in rice (Fig 2). Interestingly in rice, there is a 'dip' in variant density ~5 kb before border, but the dip corresponds to increased gene density in that regions (Fig 2). The corresponding discussion can be found on Pages 3-4, Lines 69-96 of the revised manuscript.

3. It would be good to indicate how the "random" control was done in you method.

We now include detail for random control generation (Page 10, Lines 294-295 of the revised

manuscript). In short, random controls were generated by random re-distribution of TAD boundaries across the rice genome using regioneR v1.15.2. The same script was used to generate windows around TAD boundaries for both true and randomly re-distributed boundaries.

Reviewer #2 (Remarks to the Author):

This manuscript performed a set of computational analysis integrating existing genomic resources including rice chromatin conformation map and variants across rice populations. The manuscript described a somewhat interesting observation that nucleotide variations are enriched across TAD borders, which is different from what was observed in the human. However, the manuscript is rather preliminary and does not provide sufficient novelty or biological insights into the described observation. The results are also rather fragmented and do not provide a convincing and coherent explanation of the rice-human difference. The enrichment of meiotic crossover (CO) at TADs appears to be interesting. However, a rice-human comparison for CO was not performed and hence it is unclear whether the enrichment of CO contributes the rice-human difference. Statistical tests are also missing for a number of conclusions in the manuscript.

We apologize that the initial presentation of the results was not clear. The manuscript reports two sets of results. First, we perform, side by side comparison of human and rice variation profiles at TAD border (using the same method as designed by Fudenberg et al.² for the analysis of human genome) which suggests that in contrast to the human genome, in the rice genome no pronounced reduction of variant density at TAD boundary is observed (Section titled ‘Rice and human TAD borders have different variation profiles’ and corresponding Fig 2). Having performed the comparative human-rice analysis we made an interesting observation that in rice the overall variant profile across boundaries appears highly asymmetrical. We then turned to comparison of entire rice TADs and inter-TADs (not just the border regions) and show that overall rice TADs have higher SNP density, SV density and recombination rate compared to inter-TADs (Section titled: ‘Rice TADs display increased variant density compared to inter-TADs’, ‘Rice TAD and inter-TAD regions differ in epigenetic landscape, TE density and meiotic crossover rate’ and corresponding Figs 3 and 5). We have re-structured the manuscript to emphasize that the two distinct analyses.

1. Many genomic features being analyzed in this manuscript are not clearly defined. To name a few examples, Line 34-36: It is unclear what the authors mean with “within TADs”, or “surrounding genomic sequence”. What is the size of these features? Line 36-37: what is the definition of “non-TAD” regions? Does it mean all genomic regions not annotated as TAD, or only surrounding regions of TADs?

We now make a clear distinction in the manuscript that TADs are defined by regions of high interaction strength as identified by Armatus, non-TADs (now called inter-TADs) are regions found outside of TADs (all genomic regions not annotated as TADs). We also provide corresponding size distributions (Page 3, Lines 64-69). An illustration of TAD and inter-TAD regions can be found in Fig 1A.

2. Fig. 1A top two panels: the Y-scales are too compressed that any difference between TAD and non-TAD cannot be seen easily.

Fig 1A (now Fig 3) now used log scale on Y-axis.

3. In order to compare rice to human, the authors should present the human results as part of Fig. 1B. So the reads don’t have to go back to published literature about human chromatin organizations.
We now present re-analysis of human SNP and deletion breakpoint data as part of Fig 2.

4. Fig. 1B. For the distribution of deletions and insertions, there is substantial average fluctuation of the densities surrounding TAD boundary. Is the enrichment of deletion and insertion at TAD boundary statistically significant? A shuffling experiment may be performed to test whether the amplitude of increase at TAD boundary is significant.

We apologize if the text was not clear. We did not aim to study the enrichment of variants at TAD boundary but the overall difference between TAD and non-TAD (inter-TAD) regions (tested using Wilcoxon rank sum test, Fig 3). We did note the difference in variant profiles between rice and human at TAD borders as presented in Fig 2, although these were not tested for statistical significance. For the analysis of variant profiles at TAD border we used the methods previously developed by Fudenberg et al.²

5. *Statistical tests are missing from the following lines - Line 55-56, Line 84-87, Line 90-91,* Following advice of reviewer 3 we now changed the approach for study of significant associations. We now use genome-wide correlations between rice genomic and epigenomic features (cor.mtest implemented in package corrrplot was used for significance testing ($p < 0.05$)) and overrepresentation of features positively correlated with variant density to explain accumulation of genomic variants within TADs. Where statistical tests are not specifically mentioned in the text, the necessary details can be found in the associated figure.

6. *Line 60. How are TAD and non-TAD regions defined here? What are the sizes?* We now make a clear distinction in the manuscript that TADs are defined as regions of high interaction strength as identified by Armatus, non-TADs (inter-TADs) are regions found outside of TADs (all genomic sequence not annotated as TADs). We also include an illustration in Fig 1A and size distribution for both (Page 3, Lines 63-65).

7. *The authors should perform rice-human comparison for the TE and meiotic crossover enrichment analyses to ask whether these are possible contributors for the observed rice-human differences.* The only comparative analysis between rice and human performed was regarding SNP and SV breakpoint distribution at TAD border. It has already been reported that in human systems the reduction in variant density at TAD border can be explained by requirement for sequence recognition by CTCF and has not been linked to TEs or meiotic crossover^{2,3}. We simply extend this conclusion, suggesting that lack of similar profile in rice may reflect lack of requirement of sequence recognition for TAD formation. We believe that a detailed analysis of human TEs and meiotic crossover data is out of the scope of this manuscript. We did however perform re-analysis of human SNP and SV breakpoint variation at TAD border (Fig 2), using similar method to that used for human analysis

Reviewer #3 (Remarks to the Author):

This manuscript by Agnieszka et al. asked the question whether presence of topologically associated domains or TADs in the rice genome is associated with reduced single nucleotide polymorphisms (SNPs) and structural variations as previously reported in the human genomes. Authors used the 3000 rice genome resequencing data for this purpose. They found increased SNPs and structural variant density across TAD borders and within TADs in rice, which was in contrast to findings in human genomes. Potential effects of differences in gene density between TADs and non-TADs were ruled out by separately analyzing genic and non-genic loci across the rice genome. Next, authors tried to associate the high rate of genetic variation with the TAD epigenetic landscape, TE composition and meiotic crossover rates. They concluded that epigenetic modifications, partitioning of different TE families as well as meiotic crossing rates may all contribute to the higher level of genetic variation in TADs than non-TADs in rice.

The subject of this manuscript is topical. Overall, the analytical methods are sound and the manuscript is well-written. It provides new insight into the topological chromatin structure and its maintenance in rice genome, although the study is only based on sequence-level correlative analyses and descriptions while lack of any experimental causal validation. It also remains unknown whether trends identified in rice will apply to other plant genomes even within the monocots. For example, if differential presence of TEs is a major contributing factor, then we would expect fundamental differences between rice and maize, barley and wheat. In this aspect, the study and conclusions cannot be considered as generally applicable. It only draws the attention that there is a difference between rice and human genomes with respect to the genetic mutation rate of TADs vs. non-TADs. In

addition, some of the conclusions may not have been strongly supported by even by the data and/or analyses.

We now include a detailed analysis of correlations between genomic and epigenomic features across the rice genome. We split the rice genome into 40 kb bins (similar to median size of TAD identified) and studied correlations between features. We then link the correlation observed to over-representation of features positively correlated with variant density within TADs to explain accumulation of genomic variants within TADs. Sections titled ‘Genome-wide analysis reveals patterns of correlation of genomic and epigenomic features across the rice genome’ and ‘Rice TAD and inter-TAD regions differ in epigenetic landscape, TE density and meiotic cross-over rate’ While we agree that the conclusions are not generally applicable across plant species, we believe they constitute a starting point for more detailed comparative analysis. We do note an interesting parallel between our observations and that made in *Drosophila* (Section titled: ‘Relationship between the 3D chromatin structure and sequence variation in the rice genome’).

I have the following additional comments.

1. I wonder if the cell-type-specific TADs identified by Dong et al, 2017 (Molecular Plant) is more suited for this purpose than those by Liu et al, 2017 used in this manuscript. TADs identified in the latter paper utilized the rice seedlings as its input tissue. Accordingly, the respective Hi-C interaction map was a weighted average interaction status across all various cell types, although majority should be the mesophyll cells. To avoid potential effects of the cell-type differences and ensure the specificity and accuracy of the utilized TADs, it might be better to use the pure mesophyll cell TADs characterized by Dong et al, 2017, or, to use both.

Following advice of reviewer 1, we performed full TAD discovery in all three datasets using Armatus (Supplementary text). Unfortunately, the mesophyll dataset has the lowest number of valid read pairs/interactions as identified by HiC-Pro. As a result, it is not really suitable for high resolution TAD discovery, which is reflected in the original manuscript which uses lower resolution than Liu et al. We did however perform comparison of intra-TAD interactions in all three datasets and we find that TADs discovered using Liu et al dataset, still have high levels of intra-TAD interactions in mesophyll (Supplementary text).

2. It seems to me that according to presentations in Fig.1B and Fig. S1, except for the SNP density in non-genic regions (Fig. S1 bottom right panel), the densities of “Deletion and Insertion” around the TAD borders and within the TADs (TAD inner regions) are largely (in both genic and non-genic regions) indistinguishable from those generated by random selection (set as the background). Accordingly, at Page 2 lines 54-56, it needs to be further specified as “only SNP density in non-genic region was elevated across TADs (around the TAD borders and within the TAD boundaries).”

We have now updated the manuscript with new results based on Armatus called TADs. Analysis of the variation profile across TAD boundary was used mainly for comparison with human dataset (Fig 2 of the revised manuscript). However, when entire TADs and inter-TAD regions are compared, TADs were significantly higher in SNPs, deletions and insertion break points (Fig 3 of the revised manuscript). The results have now been reorganized and can be found under two separate headings (‘Rice and human TAD borders have different sequence variation profiles’ and ‘Rice TADs display increased sequence diversity compared to inter-TAD regions’)

3. Page 2, line 59, it is indeed difficult to establish a true causal relationship between the elevated sequence variation and the TADs. What can be explore is the association between these two properties, genomic and chromatin architecture.

We now perform genome-wide analysis of correlation between different genomic and epigenomic features of the rice genome and the link the correlations observed with increased incidence of certain features in TADs. We thank the reviewer for an excellent suggestion.

4. Page 2, lines 57-58, the statement about “the lack of insulator proteins identified in plants” was inaccurate. It is true that there are no CTCF factors in plants; however, isolative cohesins are encoded in plant genomes.

The text has been amended accordingly. Page 3, Lines 74-75 of the revised manuscript.

5. Page 3, lines 64-67, I suggest authors emphasize the distinct differences of enrichment in active and repressive histone markers in TAD border and TAD inner regions, respectively. The enrichment of H3K9me2 within TAD could only explain the mutation accumulation in TAD inner regions not that around the TAD borders.

We now separate results from analysis of variant profile across TAD borders (used for comparisons with available human data) and the remainder of the results which focus on the entire TADs and inter-TAD regions.

3. Page 5, line 147, it should be “the coordinates of rice TADs were obtained from a previous publication⁵”?

We now performed TAD discovery using Armatus. The methods have been updated accordingly.

1. Forcato, M. *et al.* Comparison of computational methods for Hi-C data analysis. *Nature Methods* **14**, 679-685 (2017).
2. Fudenberg, G. & Pollard, K.S. Chromatin features constrain structural variation across evolutionary timescales. *Proceedings of the National Academy of Sciences* **116**, 2175 (2019).
3. Rocha, P.P., Raviram, R., Bonneau, R. & Skok, J.A. Breaking TADs: insights into hierarchical genome organization. *Epigenomics* **7**, 523-526 (2015).

REVIEWERS' COMMENTS:

Reviewer #1 (Remarks to the Author):

The revised manuscript is now much improved. My concerns have been reasonably addressed.

I have one last comments. Although authors have also analysed some feature of rice TAD boundary, most of case they have present data in rice for TAD and inter-TAD, while that in human is TAD and TAD boundary. What would it be, if you introduce inter-TAD in human data? In other words, would the teature of inter-TAD (about 30% of rice genome) be applied to human or other plant and animal species?

Reviewer #2 (Remarks to the Author):

The revised manuscript has addressed the specific comments I brought up during the review. However, I would have to maintain my overall assessment that the manuscript is "rather preliminary and does not provide sufficient novelty or biological insights into the described observation". Certainly, I would leave it to the editor to determine whether the manuscript is a good fit for the journal.

Reviewer #3 (Remarks to the Author):

I have carefully read the revised manuscript an authors' responses. The manuscript has been substantially improved. Also, most of my comments to the previous version have been addressed to the extent that is possible based on data analyses alone. Thus, some points, such as lack of experimental data to confirm computational analyses or to link the observations to biological meaning, seem beyond the scope of this manuscript. Although it remains unclear whether the observations apply only to rice or more generic, I agree that it serves as a start that may spur further studies concerning the subject. I do not have additional comments.